# SNX17 Recruits USP9X to Antagonize MIB1-Mediated Ubiquitination and Degradation of PCM1 during Serum-Starvation-Induced Ciliogenesis

**DOI:** 10.3390/cells8111335

**Published:** 2019-10-29

**Authors:** Pengtao Wang, Jianhong Xia, Leilei Zhang, Shaoyang Zhao, Shengbiao Li, Haiyun Wang, Shan Cheng, Heying Li, Wenguang Yin, Duanqing Pei, Xiaodong Shu

**Affiliations:** 1School of Life Sciences, University of Science and Technology of China, Hefei 230027, China; wang_pengtao@gibh.ac.cn; 2CAS Key Laboratory of Regenerative Biology, South China Institutes for Stem Cell Biology and Regenerative Medicine, Guangzhou Institutes of Biomedicine and Health, Chinese Academy of Sciences, Guangzhou 510530, China; xia_jianhong@gibh.ac.cn (J.X.); zhang_leilei@gibh.ac.cn (L.Z.); zhao_shaoyang@gibh.ac.cn (S.Z.); li_shengbiao@gibh.ac.cn (S.L.); wang_haiyun@gibh.ac.cn (H.W.); li_heying@gibh.ac.cn (H.L.); wenguang.yin@gmail.com (W.Y.); 3Guangdong Provincial Key Laboratory of Stem Cell and Regenerative Medicine, Guangzhou 510530, China; 4Hefei Institute of Stem Cell and Regenerative Medicine, Guangzhou Institutes of Biomedicine and Health, Chinese Academy of Sciences, Guangzhou 510530, China; bioshan123@163.com; 5Guangzhou Regenerative Medicine and Health Guangdong Laboratory (GRMH-GDL), Guangzhou 510005, China; 6Joint School of Life Sciences, Guangzhou Institutes of Biomedicine and Health, Chinese Academy of Sciences, Guangzhou Medical University, Guangzhou 511436, China

**Keywords:** SNX17, USP9X, PCM1, MIB1, centriolar satellite, cilia

## Abstract

Centriolar satellites are non-membrane cytoplasmic granules that deliver proteins to centrosome during centrosome biogenesis and ciliogenesis. Centriolar satellites are highly dynamic during cell cycle or ciliogenesis and how they are regulated remains largely unknown. We report here that sorting nexin 17 (SNX17) regulates the homeostasis of a subset of centriolar satellite proteins including PCM1, CEP131, and OFD1 during serum-starvation-induced ciliogenesis. Mechanistically, SNX17 recruits the deubiquitinating enzyme USP9X to antagonize the mindbomb 1 (MIB1)-induced ubiquitination and degradation of PCM1. SNX17 deficiency leads to enhanced degradation of USP9X as well as PCM1 and disrupts ciliogenesis upon serum starvation. On the other hand, SNX17 is dispensable for the homeostasis of PCM1 and USP9X in serum-containing media. These findings reveal a SNX17/USP9X mediated pathway essential for the homeostasis of centriolar satellites under serum starvation, and provide insight into the mechanism of USP9X in ciliogenesis, which may lead to a better understating of USP9X-deficiency-related human diseases such as X-linked mental retardation and neurodegenerative diseases.

## 1. Introduction

Centrosome functions as microtubule-organizing center (MTOC), which is required for the organization of mitotic spindle apparatus in mitotic cells; while in resting cells, it regulates ciliogenesis [1]. Structurally, centrosome consists of a pair of centrioles and the surrounding pericentriolar material (PCM), which is dynamically regulated throughout cell cycle. Centriolar satellites are non-membrane cytoplasmic granules located around the centrosome and they play essential roles in delivering centrosome proteins from cytoplasm to centrosome during centrosome assembly or ciliogenesis, and deficiency in centriolar satellites lead to diseases such as ciliopathies [2]. For example, inhibition of pericentriolar material 1 (PCM1), which is a major component of centriolar satellites and a scaffold protein for the assembling of centriolar satellite particles, reduces the ciliary recruitment of Rab8 and disrupts ciliogenesis [3]. In addition, PCM1 binds to the E3 ubiquitin ligase mindbomb 1 (MIB1) and restricts it to centriolar satellites, which is essential for ciliogenesis [4]. However, MIB1 is able to ubiquitinate, and thus promote the degradation of centriolar satellite proteins including PCM1 and CEP131 [4,5]; therefore, there must be mechanisms that stabilize PCM1 and centriolar satellites during ciliogenesis. So far it has been reported that under cellular stress (UV or heat shock) induced ciliogenesis, MIB1 is inactivated in a p38-independent pathway, and thus suppression of ciliogenesis by MIB1 is removed [5]. Another possibility is that PCM1 is rescued from the MIB1-induced degradation by a deubiquitinating enzyme (DUB), and USP9X (ubiquitin-specific protease 9X) is such a candidate. USP9X is a DUB that is able to regulate the integrity of centriolar satellites in mitotic cells [6,7,8], while it remains to be established whether such a mechanism also functions during serum-starvation-induced ciliogenesis.

Sorting nexin 17 (SNX17) is a sorting nexin family protein firstly identified as a P-selectin binding protein [9]. The FERM domain of SNX17 binds to the cytoplasmic sorting motif of P-selectin [10] and prevents the lysosomal degradation of P-selectin [11]. Unbiased quantitative proteomic study of membrane proteins reveals integrin family proteins as major cargos for SNX17 in Hela cells [12]. Mechanistically, SNX17 binds to the cytosolic tail of endocytosed integrins and promotes their cell surface recycling [12,13] through a unique retriever-dependent pathway [14]. Additional cargos for SNX17 have been reported and most of them belong to cell-surface proteins, and it remains to be investigated whether SNX17 functions in additional intracellular protein trafficking pathways.

Here, we report that SNX17 deficiency disrupts ciliogenesis in hTERT-RPE1 cells. Mechanistically, SNX17 recruits and stabilizes USP9X to antagonize the MIB1-induced ubiquitination and degradation of centriolar satellite proteins including PCM1 during serum starvation. However, SNX17 appears dispensable for the homeostasis of PCM1 in the presence of serum. Thus, this study reveals a stress-induced trafficking pathway for SNX17 that maintains the homeostasis of centriolar satellites during serum starvation.

## 2. Materials and Methods

### 2.1. DNA Constructs

Full-length or truncated versions of human/mouse SNX17 and USP9X were RT-PCR amplified by standard protocol and sub-cloned into the EcoRI site of pSIN-EF1-alpha-IRES-puro with GFP-Flag-tag (90505, Addgene, Watertown, MA, USA). The coding sequence of human SNX17 was also cloned into the PCR3.1-HA vector between EcoRV and BamH1 sites to generate plasmid PCR3.1-SNX17-HA. Constructs for HA-tagged wild type, K63 (a ubiquitin mutant with all lysines mutated to arginine except for K63, which can only form K63-linked polyubiquitin chain) and K48 (a ubiquitin mutant with all lysines changed to arginine except for K48, which can only form K48-mediated polyubiquitin chain) were kindly provided by Dr. Xiaofei Zhang (GIBH). All constructs were confirmed by DNA sequencing. The following primers were used in this study: human full-length SNX17 (5′-CGACGATAAGGAATTCATGCACTTTTCCATTCCCGA-3′ and 5′-TGCTCACCATGAATTCCAGATCCTCATCTCCAATG-3′); mouse full-length snx17 (5′-CGACGATAAGGAATTCATGCACTTTTCCATTCCTGAAAC-3′ and 5′-TGCTCACCATGAATTCCAGATCCTCATCTCCAAT-3′); human SNX17-PX (5′-CGACGATAAGGAATTCATGCACTTTTCCATTCCCGA-3′ and 5′-TGCTCACCATGAATTCTGTCTCCTGTTGTGCCCGA-3′); human SNX17-FERM (5′-CGACGATAAGGAATTCGAGGAAGTGTCCTTGGAAGTGC-3′ and 5′-TGCTCACCATGAATTCGAGTTTGACCATAGACTCCCGG-3′); human SNX17-ΔPX (5′-CGACGATAAGGAATTCCAGCAGGTCCCCACAGAGG-3′ and 5′-TGCTCACCATGAATTCCAGATCCTCATCTCCAATGCCC-3′); human SNX17-ΔFERM (F1: 5′-CGACGATAAGGAATTCATGCACTTTTCCATTCCCGA-3′, R1: 5′-TTACTTGATGTGGGGACCTGCT-3′; F2: 5′-TCCCCACATCAAGTAAGCTGAG T-3′, R2: 5′-TGCTCACCATGAATTCCAGATCCTCATCTCCAATGCCC-3′); human SNX17-PX+FERM (5′-CGACGATAAGGAATTCATGCACTTTTCCATTCCCGA-3′ and 5′-TGCTCACCATGAATTCGAGTTTGACCATAGACTCCCGG-3′); human full-length USP9X (5′-CGACGATAAGGAATTCATGACAGCCACGACTCGTGG-3′ and 5′-TGCTCACCATGAATTCTTGATCCTTGGTTTGAGGTGGG-3′); human USP9X-N1 (5′-CGACGATAAGGAATTCATGACAGCCACGACTCGTGG-3′ and 5′-TGCTCACCATGAATTCTTGAACTTCTTGAACATGACTATATC-3′); human USP9X-N2 (5′-CGACGATAAGGAATTCGAACGGCTTAACTTCCTTAGATTTTT-3′ and 5′-TGCTCACCATGAATTCTCAGATATCTGCTGAGCAAGACGA-3′); human USP9X-C1 (5′-CGACGATAAGGAATTCTACATTGGCACAGCAATAACTACTTG-3′ and 5′-TGCTCACCATGAATTCTGGCATAATAATCTGATGAGGTCTG-3′); human USP9X-C2 (5′-CGACGATAAGGAATTCTCAGCCATTGAGAGAAGTGTACGG-3′ and 5′-TGCTCACCATGAATTCTTGATCCTTGGTTTGAGGTGGG-3′).

### 2.2. Cell Culture, Transfection, Ciliogenesis, and Drug Treatment

HEK 293T cells were grown in Dulbecco’s modified Eagle’s medium (DMEM) (SH30022.01, Hyclone, Logan, UT, USA) supplemented with 10% fetal bovine serum (FBS) (SFBE, Natocor, Cordoba, Argentina) at 37 °C with 5% CO2. Human hTERT immortalized retina pigmented epithelial cells (hTERT-RPE1, RPE1 hereafter) were maintained in DMEM-F12 culture medium (SH30023.01, Hyclone, Watertown, MA, USA) supplemented with 10% FBS. DNA constructs were transfected into RPE1 cells using the Lipofectamine 3000 (L3000008, Life Technologies, Carlsbad, CA, USA) and siRNAs were transfected into RPE1 cells using the RNAiMAX (13778030, Life Technologies, Carlsbad, CA, USA) according to manufacturers’ protocols. The following siRNAs were used: siNC (5′-TTCTCCGAACGTGTCACGTTT-3′, negative control siRNA); siSNX17-1 (5′-GTACCATTGCCCAGTGGAA-3′); siSNX17-2 (5′-GAGTCAAGAGTATAAGATT-3′); siMIB1-1 (5′-GGATAAAGATGGTGATAGA-3′) and siMIB1-2 (5′-GAAGAAAGATGATGGTTAT-3′); siUSP9X-1 (5′-AGAAATCGCTGGTATAAAT-3′), siUSP9X-2 (5′-ACACGATGCTTTAGAATTT-3′), siUSP9X-3 (5′-GTACGACGATGTATTCTCA-3′), and siUSP9X-4 (5′-GAAATAACTTCCTACCGAA-3′). The final concentration for siNC, siSNX17-1, or siSNX17-2 was 50 nM. For MIB1 knockdown, a combination of siMIB1-1 and siMIB1-2 (25 + 25 nM final concentration) was used. For USP9X knockdown, a combination of siUSP9X-1, 2, 3, and 4 (final concentration of 5 nM for each siRNA) was used. Knockdown efficiency at protein level was determined by western blot.

For ciliogenesis experiments, RPE1 cells were cultured on coverslips to confluence, and then serum-starved for 24 or 48 h. For rescue experiments, SNX17 knockout RPE1 cells were infected with pSIN-GFP or pSIN-mSNX17-GFP lentivirus for 72 h and then serum-starved for 48 h. Cells were then fixed and processed for immunofluorescence staining. Three biological repeats were performed and at least 100 cells were analyzed for cilium formation in each group.

For drug treatment, the proteasome inhibitor MG132 (M7449, Sigma, Saint Louis, MO, USA) or the lysosome inhibitor Bafilomycin A1 (S1413, Selleck, Houston, TX, USA) was used. For analysis of PCM1 degradation, cells were treated with MG132 (1 µM) or Bafilomycin A1 (2 nM) in the absence of serum for 48 h. For analysis of USP9X degradation, cells were treated with MG132 (1 µM) or Bafilomycin A1 (2 nM) in the absence of serum for 24 h. In total, 0.1% DMSO was used as negative control in these experiments. Cells were then harvested for immunoprecipitation and western blot as described below.

### 2.3. SNX17 Knockout

SNX17 knockout RPE1 cell lines were generated using the lentiCRISPRV2 system [15] with the targeting sequence (5′-CTTTCAACAGTTTCCTGCGTCGG-3′) located at exon 2 of human SNX17 gene. Briefly, HEK 293T cells were transfected with the packaging plasmid (psPAX2), envelop plasmid (pMD2.G), and expression vector (3:1:4 ratio, 6 µg DNA in total), and cells were grown in 6-cm dishes for 2 days. Viral supernatant was then collected with 0.22-µm filter, and RPE1 cells were infected with the media containing lentivirus (1:1) and polybrene (8 µg/mL, H9268, Sigma, Saint Louis, MO, USA) for 48 h. Infected RPE1 cells were selected with puromycin (2.5 µg/mL, S7417, Selleck, Houston, TX, USA) for 48 h. Cells were then seeded as single cell into 96-well plates for 2 weeks and screened for SNX17 knockout mutants. For genotyping, genomic DNA was extracted from the infected RPE1 cells using TIANamp Genomic DNA Kit (DP304, Tiangen Biotech, Beijing, China) following the manufacturer’s instructions. PCR products spanning the sgRNA target site were generated with the LA Taq^®^ DNA Polymerase (RRO2MQ, Takara, Otsu, Japan) with primers (5′-AGGAAGGCTCTTGTTTGA-3′ and 5′-TAGGCTGTGATTTCTTTGAT-3′) and sequenced. Two independent mutants were obtained after DNA sequencing (detailed in Figure 1D) and SNX17 knockout were confirmed at protein levels by western blot.

### 2.4. Immunofluorescence Staining

RPE1 cells cultured on coverslip were washed with PBS and fixed in 4% PFA at RT for 30 min, permeabilized in 0.1% Triton X-100 (T8787, Sigma, Saint Louis, MO, USA)/PBS for 15 min, washed with PBS, and then blocked in 5% FBS/PBS for 1 h at RT. Samples were then incubated with a primary antibody (diluted in blocking solution) at 4 °C overnight, washed several times with the blocking solution, and then incubated with a secondary antibody for 2 h at RT. After several washes, DNA was counterstained with DAPI (D9564, Sigma, Saint Louis, MO, USA). Samples were then mounted with mounting solution (P36934, Life Technologies, Carlsbad, CA, USA) and imaged using the Zeiss LSM 710 confocal microscope (Carl Zeiss, Oberkochen, Germany). Primary and secondary antibodies used in this study were listed in Appendix A.

### 2.5. Immunoprecipitation and Western Blot

Wild type and mutant RPE1 cells were cultured in 100-mm plates in the presence or absence of FBS and harvested at the indicated time points. Cells were washed with PBS and then lysed in 500 µL RIPA buffer (50 mM Tris-HCl pH7.4, 150 mM NaCl, 2 mM EDTA, 1% NP-40) containing 1% protease inhibitor cocktail (04693132001, Roche, Basel, Switzerland) plus 1 mM PMSF (78830, Sigma, Saint Louis, MO, USA) for 30 min on ice. After centrifugation at 3600 rpm for 4 min, supernatants (450 µL) were collected and incubated with the PCM1 antibody (5 µg) overnight at 4 °C, and then incubated with the Protein A Resin (50 µL, L00210, Genscript, Piscataway, NJ, USA) for 4 h at 4 °C. For pull-down of Flag-tagged proteins, samples in RIPA buffer were incubated with Flag beads (20 µL, A2220, Sigma, Saint Louis, MO, USA) overnight at 4 °C. The resin was then collected by centrifugation (3600 rpm for 5 min) and washed with RIPA buffer (500 µL) for five times and ready for western blot analysis. For pull-down of HA-tagged proteins, the Magnetic Beads (20 µL, 88837, Thermo Fisher, Waltham, MA, USA) were incubated with samples overnight at 4 °C and then collected with magnetic separator. Samples were then washed and analyzed as described above.

Western blot was performed by standard protocol. Briefly, cell lysates or immunoprecipitated samples were boiled in loading buffer (50 mM Tris-HCl pH 6.8, 10% glycerol, 1% ß-mercaptoethanol, 2% SDS, and 0.1% bromophenol blue), separated by SDS-PAGE and then transferred to polyvinylidene difluoride (PVDF) membranes (IPFL00010, Millipore, Darmstadt, Germany). After blocking in 5% nonfat dry milk in TBST buffer (10 mM Tris-HCl pH 8.0, 150 mM NaCl, 0.05% Tween 20) at RT for 1 h, samples were incubated in a primary antibody at 4 °C overnight. After several washes, blots were incubated with a secondary antibody for 1 h at RT. Blots were then washed and detected with the ECL Western Blotting Detection system (WBKLS0500, Millipore, Darmstadt, Germany). Primary and secondary antibodies used in western blot were listed in Appendix A.

### 2.6. Ubiquitination Assay

To assess PCM1 ubiquitination in vivo, wild type and mutant RPE1 cells cultured in 100-mm plates were transfected with HA-tagged K63 or K48 linkage-specific ubiquitin plasmid (6 µg), respectively. After 36 h of transfection, cells were changed to serum-free media for 0, 24, and 48 h, and then harvested. Cell extracts were immunoprecipitated with the PCM1 antibody (5 µg) overnight at 4 °C, and then incubated with the Protein A Resin (50 µL) for 4 h at 4 °C. Samples were then collected and analyzed by western blot as described above.

### 2.7. Statistical Analyses

For ciliogenesis experiments, three independent biological repeats were performed and at least 100 cells were analyzed in each group. Data represent mean ± SD of the percentages of cilia positive cells calculated using the Graphpad Prism 6 (Graphpad, San Diego, CA, USA) software. Statistical differences were determined by unpaired two-tailed Student’s *t*-test between two groups, one-way ANOVA with Tukey’s multiple comparison tests as a post hoc test for comparing every mean to every other mean or one-way ANOVA with Dunnett’s Multiple Comparison as a post hoc test for comparing every mean to a control mean in each data set. A *p*-value < 0.05 was considered statistically significant.

## 3. Results

### 3.1. SNX17 Is Required for Serum-Starvation-Induced Ciliogenesis

In our previous investigation of the role of SNX17 in Notch signaling [16], we noticed that knockdown of SNX17 led to ciliopathy-related defect in zebrafish. To determine whether SNX17 is cell-autonomously required for ciliogenesis, we turn to the well-established RPE1 cell-based in vitro ciliogenesis model. We first performed loss-of-function studies. RPE1 cells were transfected with a negative control siRNA (siNC) or siRNAs to human SNX17, and both siRNAs to SNX17 effectively reduced the protein levels of SNX17 as determined by western blot (Figure 1A). Ciliogenesis was induced by serum starvation for 48 h, and cilia were visualized by immunofluorescence staining with an anti-acetylated tubulin antibody (Ac-Tub). We found that 63.5% of the siNC-treated cells contained cilia (mean of three independent biological repeats, *n* > 100 in each experiment). SNX17 knockdown reduced ciliogenesis rate to 32.6% (for siSNX17-1, *p* < 0.001) or 33.6% (for siSNX17-2, *p* < 0.01) (Figure 1B,C), indicating an essential role of SNX17 in ciliogenesis. To further validate this observation, we used CRISPR/Cas9 technology to knockout SNX17 gene in RPE1 cells and determined its effect on ciliogenesis. We obtained two mutant cell lines with both alleles of SNX17 disrupted (Figure 1D,E). We then investigated their ciliogenic capabilities upon serum starvation. We found that only 10.1% of the MU1 cells formed cilia at 48 h after serum starvation, which was significantly reduced when compared to that in WT cells (69.9%, *p* < 0.0001). Similar ciliogenic defect was observed in MU2 mutant (12.0%, *p* < 0.0001 when compared to WT) (Figure 1F,G). We then performed rescue experiments to confirm the ciliogenic defect in mutant cells was indeed induced by SNX17 deficiency. We found that lentivirus-mediated expression of GFP did not affect ciliogenesis in MU1 (11.2%, *p* = 0.99 when compared to uninfected MU1) while expression of GFP-tagged mouse SNX17 gene (mSNX17-GFP) partially restored cilium formation in MU1 (29.0%, *p* = 0.0002 when compared to uninfected MU1; *p* = 0.0003 when compared to GFP-infected MU1) (Figure 1F,G). These results clearly demonstrate that SNX17 is required for serum-starvation-induced ciliogenesis in RPE1 cells.

### 3.2. SNX17 Regulates Homeostasis of Multiple Centriolar Satellite Proteins During Ciliogenesis

SNX17 is well established as a phosphatidylinositol-3-phosphate (PI3P) binding protein [16] that regulates the recycling of endocytosed membrane proteins from endosomes to cell surface under normal culture condition. Whether or not SNX17 is involved in vesicular trafficking to centrosome or cilia remains unknown. To address this question, we first analyzed the protein level as well as the subcellular distributions of SNX17 in the presence or absence of serum. We found that the protein level of SNX17 was not affected by serum starvation (Figure 2A). Co-localization of ectopically expressed SNX17-GFP with centrosome marker pericentrin (PCNT) was detected in all cells analyzed either in the presence (*n* = 9) or absence of serum (*n* = 4) (Figure 2B). We found that very few SNX17-GFP co-localized with the centriolar satellite marker PCM1 in serum-containing media (0/4); however, co-localization of SNX17 and PCM1 was detected in all cells after 48 h of serum starvation (*n* = 5) (Figure 2B). We then determined whether SNX17 localized to ciliary membrane and found that SNX17 was not detected on mature cilia (*n* = 4) (Figure 2B). Together, these results indicate SNX17 functions in early stages of ciliogenesis.

Centrosome organizes the transporting of proteins and vesicles to initiate ciliogenesis. The centrosomal distribution of SNX17 prompted us to test whether centrosome- or ciliary vesicle (CV)-related proteins are disrupted upon SNX17 knockout. We found by western blot that the protein levels of centrosome-related proteins such as gamma-tubulin (TUBG1), PCNT, and CEP97 were not affected in the absence of SNX17 under both normal and serum-starved conditions (Figure 3A). Vesicular markers involved in ciliogenesis such as RAB8A and RAB11 also appeared normal in SNX17 mutants. However, we found that the protein levels of several centriolar satellite proteins including PCM1, CEP131, and OFD1 were clearly reduced in SNX17 mutant cells after serum starvation, while homeostasis of these proteins in the presence of serum appeared unaffected in the mutant cells (Figure 3A). We determined the mRNA levels of these genes and found they were not disrupted in SNX17 mutant cells (Appendix A), indicating a defect in post-translational regulation. Immunofluorescence staining confirmed the PCM1 defect. Co-localizations of PCM1 and TUBG1 were detected in WT cells either in serum (26/32) or serum-free (27/31) media (Figure 3B). In SNX17 mutant cells, PCM1 was detectable in 23 out of 25 cells in the presence of serum; however, PCM1 was clearly degraded after serum starvation (only 3 out of 19 cells with detectable PCM1 at centrosomal regions) (Figure 3B). We then determined whether the ciliary vesicles were disrupted in mutant cells. We found that RAB11 co-localized with TUBG1 in all WT (*n* = 38) and mutant cells (*n* = 42) analyzed (Appendix A). However, the RAB8A vesicles at the ciliary membrane were reduced in the mutant cells (30/33 in WT and only 1/9 in MU1) (Appendix A). To further characterize this defect, we performed transmission electron microscopy (TEM) analysis. We found that the formation of CVs was clearly reduced in SNX17 mutant cells at 24 h after serum starvation (Appendix A). Together, these observations suggest that SNX17 deficiency induces the degradation of a subset of centriolar satellite proteins, which disrupts downstream ciliary vesicular trafficking and results in ciliogenesis defects.

### 3.3. PCM1 Is Degraded Via the Lysosomal Pathway in SNX17 Knockout Cells

Since PCM1 is a major component and scaffold for centriolar satellites essential for ciliogenesis, we focused on the regulation of PCM1 by SNX17. To investigate the degradation pathway of PCM1, we blocked either the lysosomal degradation pathway by the proton pump inhibitor Bafilomycin A1 (Baf) or the proteasomal pathway by the inhibitor MG-132, and then tested their effects on serum-starvation-induced PCM1 degradation. We found that Baf prevented the degradation of PCM1 in SNX17 mutant cells while MG-132 failed to do so (Figure 4A). Immunofluorescence staining revealed that those Baf-rescued PCM1 co-localized with lysosome marker LAMP2 in all mutant cells examined (*n* = 17), while PCM1 in WT cells did not co-localize with LAMP2 under the same condition (0/28) (Figure 4B). These observations suggest that SNX17 prevents PCM1 entering into the lysosomal degradation pathway upon serum starvation.

Ubiquitination is a common mechanism that regulates the turnover of proteins, so we compared the ubiquitination level of PCM1 between WT and SNX17 mutant cells by western blot with the ubiquitin-specific antibody FK2. Total PCM1 protein levels in mutant cells remained comparable to that in WT cells at 24 h after serum starvation; however, total ubiquitination levels of PCM1 were dramatically increased in mutant cells at this time point (Figure 4C). Ubiquitin contains 7 lysine residues and all of them can form homotypic polyubiquitination chains. The K48-linked polyubiquitination is best characterized to promote the degradation of substrate protein via the proteasomal degradation pathway. On the other hand, K63 polyubiquitination is involved in intracellular protein trafficking and lysosomal degradation of substrate. K63 and K48 modifications can be discerned by western blot with modification-specific antibodies. We found that the K63-ubiquitination of PCM1 was clearly enhanced in SNX17 mutant cells while the K48-ubiquitination levels were less affected (Figure 4C), which was consistent with the enhanced lysosomal degradation of PCM1 in mutant cells. We then used ubiquitin mutants to further confirm the K63 ubiquitination of PCM1 in mutant cells upon serum starvation. The HA-K63 plasmid encodes a HA-tagged ubiquitin mutant with all lysine residues mutated to arginine except for the K63, and it can only form K63-linked polyubiquitin chains. Similarly, the HA-K48 is a mutant ubiquitin that can only form K48-linked polyubiquitin chains. We transfected cells with either the K63 or the K48 ubiquitin mutant, immunoprecipitated endogenous PCM1, and then determined the level of K63 or K48 ubiquitination of PCM1 by western blot with the anti-HA antibody (Figure 4D). Consistent with our previous observations, we found that the K63 but not the K48 ubiquitination of PCM1 was stimulated at 24 h after serum starvation. Together, these results indicate that the K63-ubiquitination level of PCM1 is elevated in mutant cells upon serum starvation, and those K63-ubiquitinated PCM1 is likely degraded via the lysosomal pathway.

### 3.4. Knockdown of MIB1 Rescues PCM1 and Ciliogenesis Defects in SNX17 Knockout Cells

We next investigated how ubiquitination level of PCM1 is enhanced in SNX17 mutant cells. MIB1 is a centriolar-satellite-distributed E3 ubiquitin ligase known to ubiquitinate PCM1; so we tested whether it is involved in the SNX17 regulation of PCM1. We knocked-down MIB1 by siRNA in WT and SNX17 mutant cells (Figure 5A) and measured the ubiquitination levels of PCM1 at 24 h after serum starvation. We found that knockdown of MIB1 was able to reverse the elevated ubiquitination levels of PCM1 in mutant cells as determined by the FK2 antibody. Consistent with our previous observation (Figure 4C), MIB1 affected the K63 but not K48 ubiquitination of PCM1 in SNX17 mutant cells (Figure 5A). Furthermore, the enhanced degradation of PCM1 at 48 h after serum starvation in mutant cells was rescued by siRNAs to MIB1 as determined by western blot (Figure 5B) or immunofluorescence staining (co-localization ratio of PCM1 with TUBG1 is 3/31 for siNC and 26/28 for siMIB1-treated cells) (Figure 5C). We then determined whether this rescue of centrosomal PCM1 is able to restore ciliogenesis in mutant cells. We found that knockdown of MIB1 in WT cells did not significantly change the percentages of cilia positive cells (63.1% for siNC versus 67.7% for siMIB1, *p* = 0.49); however, knockdown of MIB1 in SNX17 mutant cells restored cilia formation from 11.4% (for siNC) to 31.1% (for siMIB1, *p* < 0.01 when compared to siNC) (Figure 5D). Thus, MIB1 activity is required for the elevated ubiquitination level and degradation of PCM1 and inhibition of MIB1 is able to stabilize PCM1 and rescue cil iogenesis in SNX17 mutant cells.

### 3.5. SNX17 Recruits USP9X to Antagonize the MIB1-Induced Degradation of PCM1

To investigate the molecular mechanism of SNX17 regulation of PCM1, we performed pull-down assay to determine whether SNX17 is able to co-immunoprecipitate with PCM1. We found that SNX17 was able to pull-down PCM1, which was greatly enhanced by serum starvation (Figure 6A). Consistent with this result, we found that co-localization of SNX17 with PCM1 was readily detected after serum starvation (Figure 2B). To further characterize this interaction, we generated truncated versions of SNX17 and determined their abilities to pull-down PCM1. We found that both the PX and the FERM domain of SNX17 were required while the C-terminal end of SNX17 was dispensable for the interaction with PCM1 (Figure 6B). We then tested whether MIB1 is present in the immune-complex and found that MIB1 did not co-immunoprecipitate with SNX17 under the same condition (Figure 6A). These data indicate that SNX17 may modulate the ubiquitination level of PCM1 by recruiting effector proteins other than MIB1.

The ubiquitination level of a target protein is balanced by ubiquitinating and deubiquitinating enzymes. We investigated whether SNX17 is able to recruit a DUB to counteract the MIB1-induced ubiquitination of PCM1. USP9X is a promising candidate as it is recently identified as a DUB for PCM1 in mitotic cells [6,7,8] and it plays a favorable role in serum-starvation-induced ciliogenesis [17]. Indeed, we found that SNX17 was able to pull-down USP9X either in the presence or absence of serum (Figure 6A, bottom panel) and co-localization of them was detected as well (Appendix A). We then analyzed the protein domains in SNX17 required for the interaction with USP9X and found that truncated SNX17 protein with either the PX domain or the FERM domain removed failed to pull-down USP9X (Figure 6C, right panel), indicating that both the PX and the FERM domains are required for the interaction. Similar truncation study revealed that the C-terminal fragment of USP9X (the C2 in Figure 6C) is sufficient to pull-down SNX17 (Figure 6C, left panel). These data indicate that the PX-FERM domain of SNX17 is involved in the recruitment of USP9X as well as PCM1. This raised the question whether PCM1 and USP9X compete with each other for SNX17 binding. We tested this possibility by measuring the ability of SNX17 to pull-down endogenous PCM1 in the increasing amounts of USP9X. We found that ectopically expressed USP9X did not reduce the pull-down of PCM1 by SNX17 (Appendix A), indicating SNX17 might bind to both proteins, and thus bringing USP9X to PCM1. We then determined whether USP9X regulates the ubiquitination and total protein level of PCM1. We found that siRNA knockdown of USP9X did not induce the degradation of PCM1 in the presence of serum; on the other hand, PCM1 protein in these cells was degraded upon serum starvation (Appendix A). We then determined the ubiquitination of PCM1 and found that the total ubiquitination level of PCM1 was elevated in USP9X knockdown cells and it was likely due to enhanced K63 but not K48 ubiquitination of PCM1 (Appendix A). Together, these data suggest that SNX17 recruits USP9X to PCM1 to remove the K63 ubiquitination of PCM1, thus preventing the lysosomal degradation of PCM1.

### 3.6. SNX17 Regulates USP9X Homeostasis Upon Serum Starvation

We next investigated whether SNX17 deficiency has an undesirable effect on USP9X. Co-localization of USP9X and PCM1 were detected in WT cells in the presence (21/22) or absence of serum (18/23) (Figure 7A). Meanwhile, this co-localization in mutant cells was 19/23 in serum-containing media while it was not detected in serum-free media (*n* = 48) (Figure 7A). We noticed that USP9X protein was detectable in mutant cells in locations other than centriolar satellites (bottom panel, Figure 7A), so we determined the total USP9X protein level by western blot. We found that the USP9X protein levels were comparable in WT and SNX17 mutant cells in serum-containing media and serum starvation had no effect on USP9X in WT cells; however, USP9X protein level in mutant was clearly reduced upon serum starvation (Figure 7B). Interestingly, the enhanced degradation of USP9X in mutant cells can be rescued by the lysosomal pathway inhibitor Bafilomycin A1 (Figure 7B,C). Together, our data suggest a model that during serum starvation, SNX17 re-localizes to centriolar satellites where it recruits USP9X to antagonize the MIB1-induced ubiquitination and degradation of PCM1; PCM1 maintains the integrity of additional centriolar satellite proteins and coordinates centrosomal and vesicular trafficking events required for ciliogenesis. In the absence of SNX17, centriolar satellite fraction of USP9X is degraded via lysosomal pathway and this leads to destabilization of PCM1 and other centriolar satellite proteins, which disrupts serum-starvation-induced ciliogenesis.

## 4. Discussion

SNX17 together with SNX27 and SNX31 consist the SNX-FERM subfamily of SNX in that they contain an atypical FERM-like domain at the C-terminal of PX domain [18]. Among them, both SNX17 and SNX27 are well studied to regulate cell surface recycling of internalized membrane proteins. In addition, SNX27 knockout is reported to lead to ciliogenesis defect in ependymal cells and hydrocephalus; however, the role of SNX27 in ependymal ciliogenesis appears to be indirect: SNX27 deficiency results in over-activation of Notch signaling and suppression of ependymal cell differentiation, which leads to decreased ependymal cell number and concomitant reduced cilium number [19]. In this study, we report that SNX17 is also required for ciliogenesis but by a cell autonomous mechanism: SNX17 recruits USP9X to maintain the integrity of centriolar satellites by antagonizing MIB1-induced degradation of PCM1. In addition to SNX17, SNX10 is the only other member of SNX family proteins that is reported to be directly involved in ciliogenesis. SNX10 does not have a FERM domain and it is unlikely to be able to recruit USP9X since both the PX and the FERM domain are required for interaction with USP9X. Indeed, SNX10 regulates ciliogenesis through the centrosomal recruitment of V-ATPase and regulates subsequent ciliary vesicular trafficking [20]. Although the underlying mechanisms differ, both studies highlight that under serum-starvation-induced ciliogenesis, SNX family proteins can regulate protein trafficking to location other than cell surface such as centriolar satellites or centrosome, which is largely ignored in previous studies. Thus, these studies provide novel insights into the mechanism and function of SNX family proteins and indicate that more sophisticated investigations of SNX family members under stressed or diseased conditions are warranted.

PCM1 is a scaffold protein essential for the integrity of centriolar satellites and it is involved in centrosome biogenesis during mitosis. In resting cells, PCM1 is required for ciliogenesis at least in part by sequestrating MIB1 in the centriolar satellites [4]. On the other hand, MIB1 is known to ubiquitinate PCM1 and overexpression of MIB1 inhibits ciliogenesis. Thus, the ubiquitination levels of PCM1 at centriolar satellites must be well-controlled during ciliogenesis. In heat shock or UV-treatment-induced ciliogenesis, MIB1 is inactivated by a p38-independent pathway, which leads to stabilization of PCM1 and ciliogenesis [5]. Such a mechanism has not been validated under serum-starvation-induced ciliogenesis. Our study suggests a different homeostatic mechanism for PCM1 during serum starvation: serum starvation induces redistribution of MIB1 to centriolar satellites and promotes PCM1 ubiquitination, which is counteracted by SNX17-mediated recruitment of USP9X to the same complex. Such a duo regulation of PCM1 by MIB1 and USP9X mirrors the situation of survival motor neuron (SMN) protein, which is ubiquitinated by MIB1 and deubiquitinated by USP9X [21,22]. In the absence of SNX17, centriolar satellite USP9X is degraded, which then leads to elevated ubiquitination level and lysosomal degradation of PCM1.

MIB1 is an E3 ubiquitin ligase known to induce K48 [23,24] as well as K63 [24,25] linked polyubiquitination of substrate proteins. As far as PCM1 is concerned, Douanne et al. reported a CYLD/MIB1 mediated degradation pathway for PCM1, which is blocked by the proteasome inhibitor MG132 [26], indicating a K48-polyubiquitination-related proteasomal degradation pathway. However, the lysosomal degradation pathway was not evaluated in that study. Here, we provided direct evidence that the K63 but not K48 ubiquitination of PCM1 was enhanced in SNX17 mutant cells during serum starvation, and inhibition of the lysosomal but not proteasomal degradation rescued the PCM1 protein in mutant cells. We further proved that knockdown of MIB1 reduced the K63 ubiquitination and degradation of PCM1 and was able to rescue the ciliogenesis defect in SNX17 mutant cells. Thus, this study highlights an important role for K63 modification in homeostasis of PCM1.

USP9X was initially identified as a DUB for Survivin and it is required for proper chromosome segregation during mitosis [27]. Further study reveals that USP9X deficiency reduces the function of spindle assembly checkpoint and enhances chromosomal instability [28]. On the other hand, USP9X overexpression promotes centrosome amplification and carcinogenesis in breast carcinomas [6] and resistance to mitotic-spindle-poison-containing chemotherapy in human B-cell lymphoma [29]. In addition to its roles in mitosis and carcinogenesis, USP9X is known to be required for neural development, and USP9X deficiency is associated with X-linked intellectual disability in patients [30,31]. Interestingly, some of the patients show symptoms of ciliopathy syndromes, while the underlying mechanisms remain to be identified [31]. A recent study identified USP9X as a DUB for PCM1, and it is required for the integrity of centriolar satellites in normal culture condition [8]. We revealed here that USP9X regulated the ubiquitination and homeostasis of PCM1 during serum starvation (Appendix A). Furthermore, USP9X is recently reported to antagonize BBS11-induced ubiquitination of NPHP5, which is required for ciliogenesis in retinal and renal cells [17]. All these studies highlight critical roles of USP9X in the homeostasis of centriolar satellite proteins, centrosome- or ciliogenesis-related proteins. Our study provides additional insight into the story: the regulation of centriolar satellite USP9X itself under serum starvation. We found that USP9X is present in centriolar satellites under normal culture condition; however, serum starvation induces elevated degradation of centriolar satellite USP9X in SNX17 mutant cells. It appears that SNX17 can bind to USP9X and rescue it from serum-starvation-induced degradation pathway. While the mechanism of serum-starvation-induced centriolar satellite USP9X degradation is currently unknown, our discovery may act as an entry point to investigate this question. Further mechanistic insight into the role of SNX17/USP9X in centriolar satellites, centrosome, and ciliogenesis may lead to a better understanding of those USP9X-deficiency-related human diseases.

## Figures and Tables

**Figure 1 cells-08-01335-f001:**
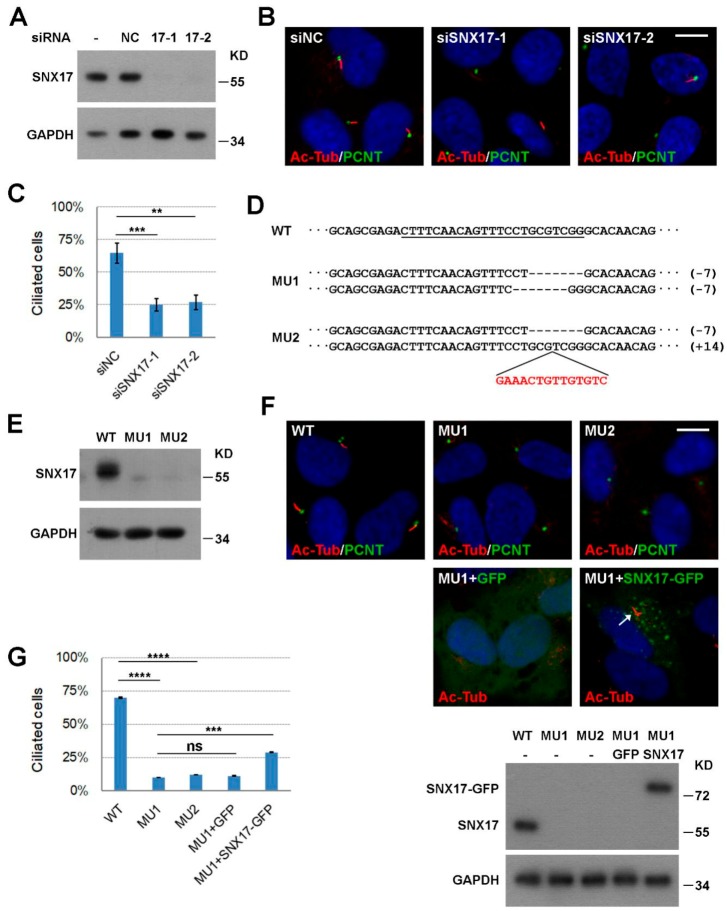
SNX17 is required for ciliogenesis. (**A**) Efficiency of siRNA-mediated knockdown of SNX17 in RPE1 cells as determined by western blot. NC is a negative control siRNA and 17-1 and 17-2 are two siRNAs to SNX17. GAPDH is loading control. (**B**) Representative immunofluorescence staining of cilia in siRNA-treated RPE1 cells at 48 h after serum starvation. Pericentrin (PCNT) is a centrosome marker and acetylated tubulin (Ac-Tub) labels cilia. (**C**) Statistical results of B. At least, one hundred cells were counted for ciliogenesis for each condition and three independent biological repeats were performed. Data represent mean ± SD from three independent biological repeats (** *p* < 0.01; *** *p* < 0.001 in one-way ANOVA with Dunnett’s multiple comparison). (**D**) CRISPR/Cas9 mediated knockout of SNX17 in RPE1 cells. The sgRNA targeting sequence in exon 2 of SNX17 gene is underlined. Two non-sense mutant cell lines with both alleles disrupted were recovered. (**E**) Western blot analysis for the expression of SNX17 in WT and mutant cell lines. (**F**) Cilium formation in WT and mutant cells at 48 h after serum starvation. Assays were performed as in B. The expression level of mouse SNX17-GFP is comparable to the endogenous SNX17 as determined by western blot using the SNX17 antibody (bottom panel). Lentivirus-mediated expression of SNX17-GFP but not GFP partially rescued cilia defect in MU1 cells (arrows). (**G**) Statistical results of (**F**). Data represent mean ± SD from three independent biological repeats (ns, non-significant; *** *p* < 0.001; **** *p* < 0.0001 in one-way ANOVA with Tukey’s multiple comparison test). Scale bar, 10 µm.

**Figure 2 cells-08-01335-f002:**
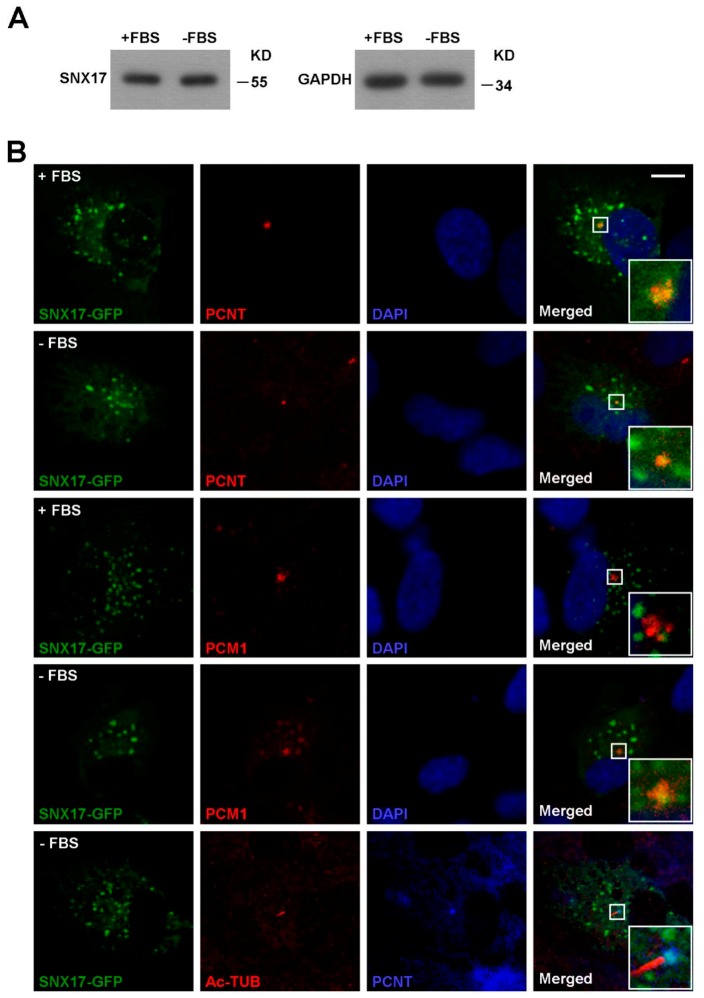
Homeostasis and subcellular distribution of SNX17. (**A**) Western blot analysis for total SNX17 protein level in the presence or absence (48 h) of serum. GAPDH was loading control. (**B**) Distribution of SNX17-GFP in RPE1 cells. Cells were transfected with a plasmid-encoding SNX17-GFP, and co-localization of SNX17 with endogenous centrosome marker PCNT or centriolar satellite marker pericentriolar material 1 (PCM1) was determined by immunofluorescence staining. AC-Tub labels the mature cilium. Scale bar, 10 µm.

**Figure 3 cells-08-01335-f003:**
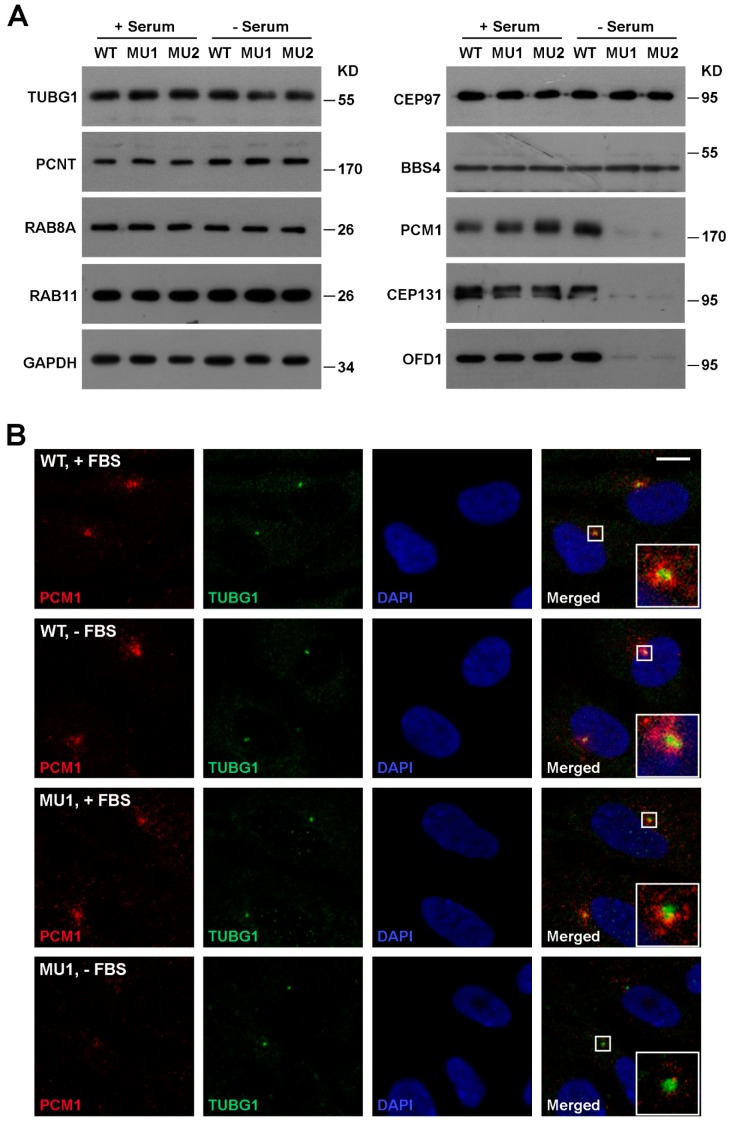
SNX17 deficiency induces degradation of multiple centriolar satellite proteins. (**A**) Western blot analysis of the indicated proteins in total cell lysate of WT or SNX17 mutant cells. Cells were cultured in either serum-containing or serum-free condition for 48 h, and then harvested for analysis. Gamma-tubulin (TUBG1), PCNT, and CEP97 are centrosome proteins. RAB8A and RAB11 are RAB family vesicular trafficking proteins involved in ciliogenesis. BBS4 is a member of the BBSsome complex that regulates the ciliary entry of RABs. PCM1, CEP131, and OFD1 are centriolar satellite proteins involved in ciliogenesis. (**B**) Distributions of PCM1 and TUBG1 in WT and mutant cells in the presence or absence of serum for 48 h. Scale bar, 10 µm.

**Figure 4 cells-08-01335-f004:**
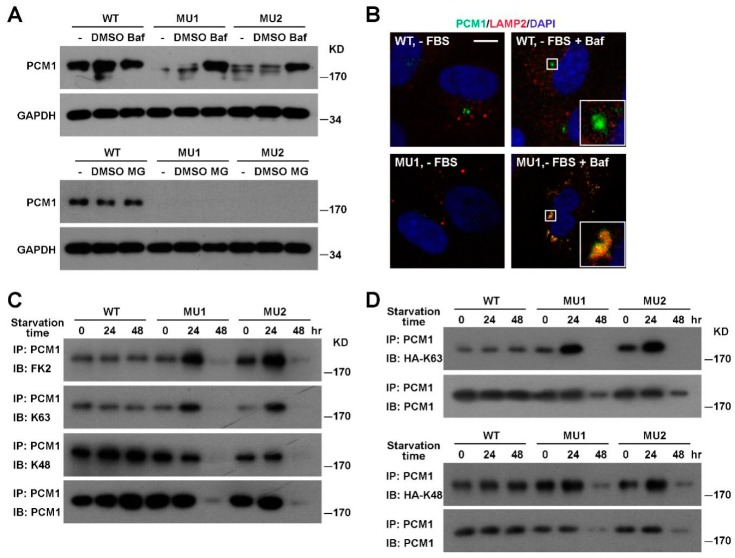
Enhanced K63 ubiquitination and lysosomal degradation of PCM1 in SNX17 mutant cells. (**A**) Pretreatment of mutant cells with the vacuolar-type H+-ATPase inhibitor Bafilomycin A1 (Baf) but not proteasome inhibitor MG-132 (MG) blocked the serum-starvation-induced degradation of PCM1. (**B**) Co-localization of PCM1 with the lysosomal marker LAMP2 was only detected in the presence of Baf in serum-starved mutant cells. Scale bar: 10 µm. (**C**) Analysis of ubiquitination levels of PCM1 in mutant cells under serum starvation. Monoclonal antibody FK2 recognizes polyubiquitinated proteins while K63- or K48-specific antibodies recognize K63- or K48-linked polyubiquitinated proteins, respectively. K63- but not K48-ubiquitinated PCM1 was accumulated in mutant cells at 24 h after serum starvation. Total PCM1 protein levels in mutant cell lysates at 24 h were comparable to that in WT cells. (**D**) Analysis of PCM1 ubiquitination by K63 or K48 linkage-specific ubiquitin mutant. HA-tagged ubiquitin mutant that can only form K63- or K48-linked ubiquitination were transfected into cells and endogenous PCM1 was immunoprecipitated at the indicated time points and the ubiquitination level of PCM1 determined by western blotting with the anti-HA antibody.

**Figure 5 cells-08-01335-f005:**
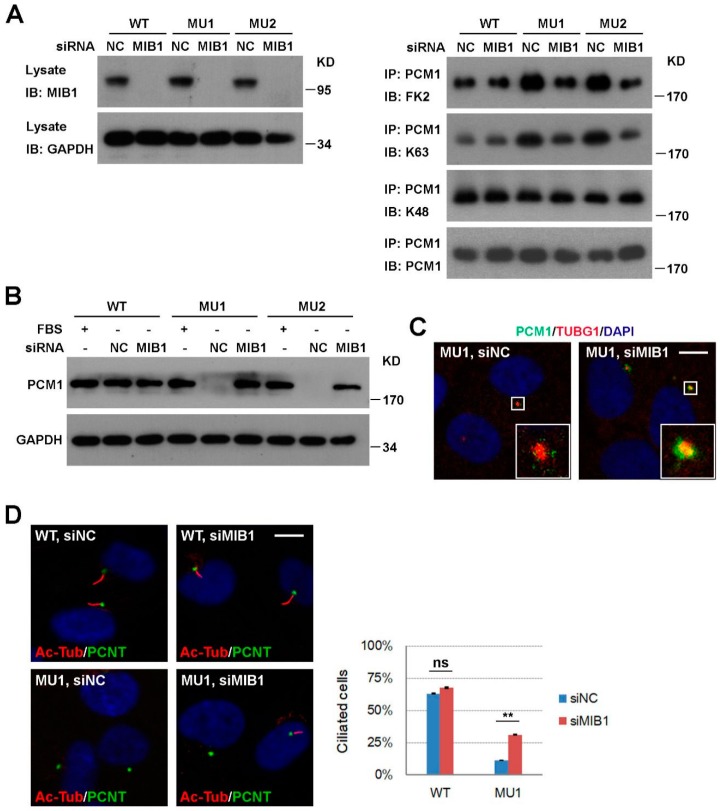
Knockdown of mindbomb 1 (MIB1) rescues ciliogenesis defects in SNX17 mutant cells. (**A**) Analysis of PCM1 ubiquitination after MIB1 knockdown. Cells were transfected with the indicated siRNAs for 48 h, serum-starved for 24 h, and then analyzed as described in Figure 4C. siMIB1 treatment rescued PCM1 protein levels in SNX17 mutant cells at 48 h after serum starvation as determined by (**B**) western blot or (**C**) immunofluorescence staining. (**D**) siMIB1 treatment restored serum-starvation-induced ciliogenesis in SNX17 mutant cells. Right panel shows the statistical results. Data represent mean ± SD from three independent biological repeats (ns, non-significant; ** *p* < 0.01 in one-way ANOVA with Tukey’s multiple comparison test). Scale bar, 10 µm.

**Figure 6 cells-08-01335-f006:**
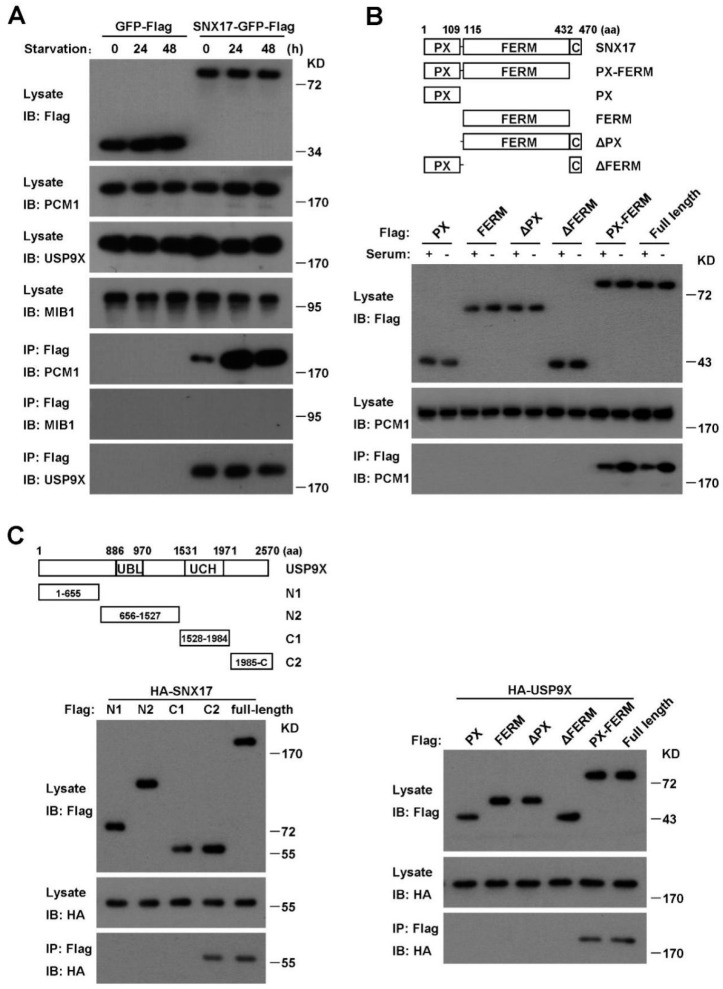
SNX17 co-immunoprecipitates with PCM1 and USP9X. (**A**) SNX17 pull-down assay. Co-immunoprecipitation of SNX17 with PCM1 was stimulated by serum starvation. SNX17 pulled-down USP9X efficiently in both serum-containing and serum-free conditions. SNX17 failed to pull-down MIB1 under the same condition. (**B**) Mapping the domains in SNX17 involved in interaction with PCM1. Deletion of the PX domain or the FERM domain but not the C-terminal fragment (aa 433–470) of SNX17 disrupted the interaction between SNX17 and PCM1. (**C**) Mapping the domains in USP9X required for interaction with SNX17. The C2 fragment of USP9X is sufficient for interaction with SNX17 while both the PX and the FERM domain in SNX17 are indispensable for interaction with USP9X.

**Figure 7 cells-08-01335-f007:**
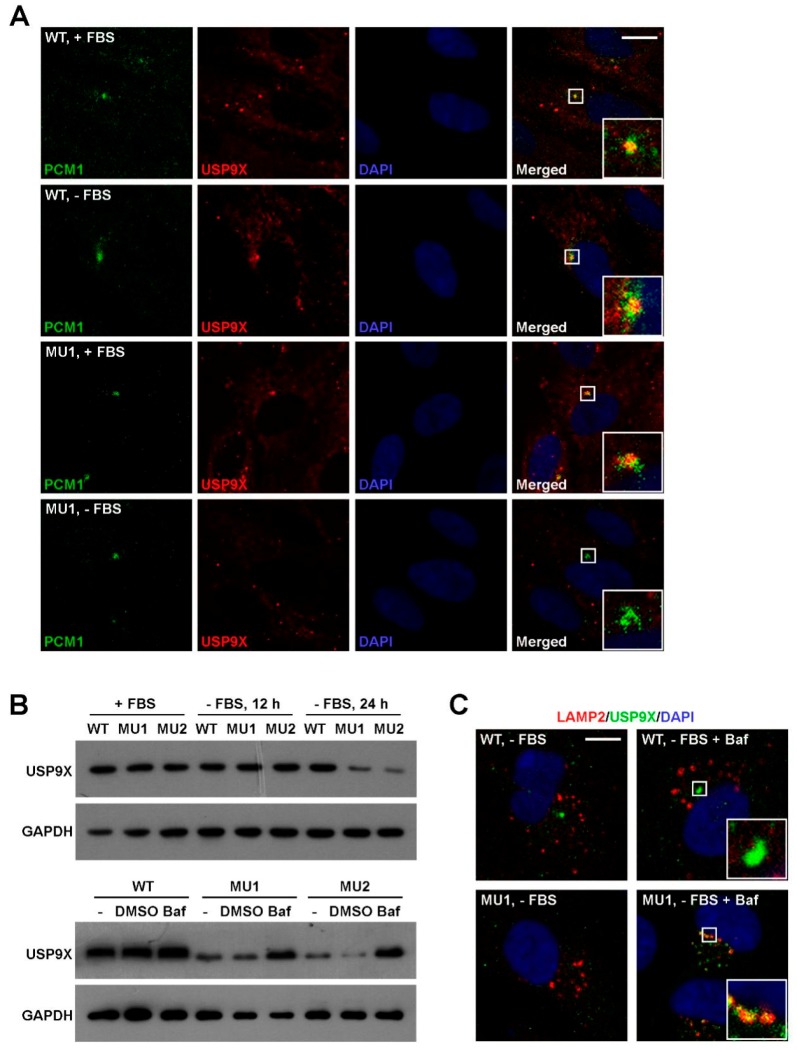
SNX17 is required for homeostasis of USP9X during serum starvation. (**A**) Co-localization of PCM1 and USP9X in WT and SNX17 mutant cells in the presence or absence of serum. (**B**) Enhanced USP9X degradation in SNX17 mutant cells at 24 h after serum starvation determined by western blot. Treatment of cells with Bafilomycin A1 (Baf) was able to rescue the elevated degradation of USP9X in mutant cells. GAPDH was loading control. (**C**) After Baf treatment, USP9X proteins were accumulated in the LAMP2-positive lysosomes in SNX17 mutant cells (15/26) but not WT cells (0/24). Scale bar in A and C: 10 µm.

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
