# Peer review of "SNX17 Recruits USP9X to Antagonize MIB1-Mediated Ubiquitination and Degradation of PCM1 during Serum-Starvation-Induced Ciliogenesis"

_cells, 2019, doi:10.3390/cells8111335_

Round 1

Reviewer 1 Report

In this study, the authors demonstrated a role of SNX17 in regulating the homeostasis of centriolar satellite proteins during ciliogenesis. Overall this study is novel and the results are interesting, but this reviewer has several concerns that need to be properly addressed.

Page 1 line 37. “While in resting cell it matures into basal body....”

Mother centriole matures into basal body, not centrosome.

Page 6 line 230. “After 48 hours of serum-starvation, SNX17 was enriched to the centrosomal region.”

To this reviewer SNX17 enrichment to the centrosomal region (Fig 2A) occurs in both serum-fed and serum-starved cells, but not specifically to serum-starved cells. Co-staining of SNX17 with a satellite marker would help.

    3. Page 6, line 254. The authors state that SNX17 deficiency disrupts vesicular trafficking. This should be further confirmed by examining Rab8a and Rab11 staining.

Fig 2B. This reviewer is wondering why BBS4 is not reduced in SNX17 mutant cells, since BBS4 is also a centriolar satellite component (Kim et al Nat Genet 2004)? Also, a previously study has shown a reduction in CEP97 in serum-starved cells (Spektor et al Cell 2007), which is in contrast to the authors’ data (compare WT + and -serum). Please explain the discrepancy. Page 9, line 317. “We found that knockdown of MIB1 was able to reverse the elevated ubiquitination levels of PCM1 in mutant cells”

Since the authors found that K63 ubiquitination of PCM1 is enhanced in SNX17 mutant cells (Fig 3), does this imply that MIB1 is the enzyme responsible for K63 ubiquitinating PCM1? Can the authors explain their results in light of the model presented by Douanne et al (Cell Rep 2019) which shows that MIB1 K48 ubiquitinates PCM1?

Author Response

Reviewer 1:

In this study, the authors demonstrated a role of SNX17 in regulating the homeostasis of centriolar satellite proteins during ciliogenesis. Overall this study is novel and the results are interesting, but this reviewer has several concerns that need to be properly addressed.

Page 1 line 37. “While in resting cell it matures into basal body....”

Mother centriole matures into basal body, not centrosome.

Our response: We agree with the reviewer’s comment and have revised the text in the resubmission.

Page 6 line 230. “After 48 hours of serum-starvation, SNX17 was enriched to the centrosomal region.”

To this reviewer SNX17 enrichment to the centrosomal region (Fig 2A) occurs in both serum-fed and serum-starved cells, but not specifically to serum-starved cells. Co-staining of SNX17 with a satellite marker would help.

Our response: We have performed co-stain of SNX17 with PCM1 and the result was presented in the revised Figure 2B. The result revealed that SNX17 co-localized with PCNT in serum-fed as well as serum-starved cell. However, few SNX17 co-localized with PCM1 in the presence of serum while serum-starvation induced their co-localization (Line 242-246 and Figure 2B of the revised manuscript).

Page 6, line 254. The authors state that SNX17 deficiency disrupts vesicular trafficking. This should be further confirmed by examining Rab8a and Rab11 staining.

Our response: We determined the distribution of RAB11 and found it co-localized with TUBG1 in WT and mutant cells (Supplementary Figure S2). We then analyzed the distribution of RAB8A and found that it was present at the ciliary membrane in WT cells (30 out of 33 cilium examined). In the SNX17 mutant cells, the formation of cilium was reduced. In mutant cells that do form cilium, most of them (8 out of 9) were RAB8A negative (Supplementary Figure S2). Together with the TEM analysis (Supplementary Figure 3), these data indicate that vesicular trafficking involved in ciliogenesis is disrupted in SNX17 mutant.

Fig 2B. This reviewer is wondering why BBS4 is not reduced in SNX17 mutant cells, since BBS4 is also a centriolar satellite component (Kim et al Nat Genet 2004)? Also, a previously study has shown a reduction in CEP97 in serum-starved cells (Spektor et al Cell 2007), which is in contrast to the authors’ data (compare WT + and -serum). Please explain the discrepancy.

Our response: Kim et al. identified PCM1 as one of the BBS4 binding proteins and demonstrated that BBS4 regulates the intracellular distribution of PCM1 (Nat Genet 2004). Meanwhile, Stowe et al. reported that siRNA knockdown of PCM1 does not disrupt the ciliary distribution of BBS4 (PMID: 22767577, Figure 2). These results indicate that PCM1 is a downstream effector of BBS4, which is consistent with our observation. In addition to BBS4, we found that the protein level of BBS7 in total cell lysate is not affected by SNX17 knockout (data not shown). These results indicate that SNX17 deficiency induces the degradation of a subset of centriolar satellite components while other satellite proteins such as BBS4 are not disrupted.

Spektor et al. reported that the CEP97 protein level is reduced in serum-starved T98G cell (Cell 2007) which is a glioblastoma-derived tumor cell line while it remains to be investigated whether the same occurs in other cell lines. In this study, we did not observed down-regulation of CEP97 in RPE1 cell (an immortalized retinal pigment epithelial cell line) after serum-starvation. The discrepancy is likely due to different regulatory mechanisms of CEP97 in different cell lines.

Page 9, line 317. “We found that knockdown of MIB1 was able to reverse the elevated ubiquitination levels of PCM1 in mutant cells”

Since the authors found that K63 ubiquitination of PCM1 is enhanced in SNX17 mutant cells (Fig 3), does this imply that MIB1 is the enzyme responsible for K63 ubiquitinating PCM1? Can the authors explain their results in light of the model presented by Douanne et al (Cell Rep 2019) which shows that MIB1 K48 ubiquitinates PCM1?

Our response: We investigated ubiquitin modification of PCM1 in SNX17 mutant cells by two different techniques (K48/K63 specific antibodies and K48/K63 specific ubiquitin mutants) and both results indicate that changes in K63 ubiquitination of PCM1 is more prominent than K48 modification in mutant cells (Figure 4C and D). We now provided additional analysis of PCM1 ubiquitination and found that knockdown of MIB1 reduced the total ubiquitination as well as the K63 ubiquitination of PCM1while the K48 ubiquitination of PCM1 is less affected (Figure 5A). These results indicate that MIB1 is likely the E3 ligase for the K63 ubiquitination of PCM1.

In a recent study, Douanne et al. convincingly showed that CYLD regulates the K63 ubiquitination and homeostasis of MIB1. The authors further reported that CYLD deficiency induces degradation of PCM1 which is partially rescued by proteasome inhibitor MG132 (Figure 3F, Cell Rep 2019), indicating a K48 ubiquitination and proteasome degradation pathway for PCM1. However, the authors did not analyze PCM1 ubiquitination directly. The authors did not test the possible function of lysosomal degradation pathway inhibitor such as Bafilomycin A1, thus this result cannot rule out the possibility that K63 ubiquitination play essential role in the homeostasis of PCM1. Indeed, we noticed that, in Figure 3F of the above-mentioned reference, MG132 treatment increases PCM1 protein level in control siRNA treatment as well, indicating the proteasome degradation pathway functions in the presence as well as the absence of CYLD. Thus, the CYLD deficiency induced degradation of PCM1 remains to be investigated.

K48 and K63 ubiquitination are not mutually exclusive. In addition to induce K48 ubiquitination, MIB1 is also reported to induce K63 ubiquitination of substrate such as BLM (PMID: 30044990) and TBK1 (PMID: 21903422). We demonstrated here that, in SNX17 mutant cells under serum starvation, K63 ubiquitination of PCM1 was clearly induced which can be reversed by knockdown of MIB1. Furthermore, knockdown of MIB1 or inhibition of the lysosomal degradation pathway rescued PCM1 protein. Our results highlighted a MIB1 induced K63 ubiquitination and lysosomal degradation pathway in the regulation of PCM1, at least during serum-starvation induced ciliogenesis in RPE1 cells.

Reviewer 2 Report

Ciliogenesis requires the concerted activity of multiple cellular pathways to construct this important organelle.  Hundreds of proteins are required and coordination of transport and stability is a key mode of regulation.  The manuscript by Wang and colleagues focuses on the ciliogenic role of SNX17, one of a family of sorting nexins that function in protein trafficking within the endosomal network.  They document a striking loss of centriolar satellite markers (including PCM1) upon serum starvation in SNX17 knockout cells.  They further implicate MIB1 as being the responsible E3 ligase for targeting PCM1, but in absence of a physical MIB1-SNX17 interaction, they turned their focus to USP9X, a deubiquitinase known to target centriolar substrates.  They document and map the physical interaction of SNX17 and USP9X, and propose a plausible model that SNX17 helps recruit USP9X to centriolar satellites upon serum starvation to offset MIB1 activity and stabilize PCM1.

Although the initial reason to focus on SNX17 is not stated, this work is interesting and should be published in a revised form, pending clarifications and additional controls.  Although many works have been published on proteolytic control at centrosomes, few have addressed how this control is linked to the vesicular trafficking network, making this work timely and important.

I have the following concerns/suggestions for the authors’ consideration:

SNX17 antibody and localization:

Although a western suggests that SNX17 antibody is specific (Fig 1A; 1E), the antibody is also used in immunofluorescence (Fig 2A).  Authors should validate the specificity of the antibody in IF using the SNX17-KO cells.

Rescue:

Using the SNX17 antibody, authors should provide a western showing levels of SNX17-GFP is the rescue experiments shown in Fig 1F-1G.

SNX17 in fed versus starved cells:

In Fig 2B, authors show that PCM1/CEP131/OFD1 etc are unstable upon cell starvation (also USP9X in Fig 6B).  Although IF images in Fig 2A suggest that SNX17 levels do not change much upon cell starvation, a western should also be provided to support this.

Overlay images:

Throughout the manuscript, many 3-color overlay images are shown for IF experiments.  It would be useful and recommended to also show grayscale, single-channel images, or channel-separated/zoomed insets (e.g. Fig 6A), especially in cases where co-localization is suggested (e.g. Fig 1F; 2A, 3B, 3D, 4C, 6C).

Data points/P-values/graphs/tables:

P-values were mentioned several times in text, without supporting data in tabular or graphic form.  Authors should attempt to address this wherever possible.

Specific examples:

Fig 2C. In text, there is mention of p<0.002 in comparing PCM1 in WT versus SNX17-KO cells in starvation. Unless data is provided as a graph and detail given of how data was analyzed/quantitated, the mention of p-value does not seem appropriate.  Counts and raw data can be provided as supplementary or extended data to support the images in Fig 2C.

Line 375-379:  Colocalization of USP9X and PCM1 were detected in WT cells in the presence (84.6%) or absence of serum (84.4%, average of three biological repeats, N>50 in each repeat) (Figure 6A). Meanwhile, this co-localization in mutant cells were reduced from 83.0% in serum-containing media to 9.0% in serum-free media (p<0.0001) (Figure 6A).

Supp FigS2: TEM analysis and counting analysis of centrosomes, ciliary vesicles and cilia in WT or SNX17-KO cells.  Since few examples were counted, it would be useful to provide more images of centrosomes and ciliary vesicles identified (not just single examples)  The text mentions “p=0.0115”, however no analysis is shown in figure or mentioned in figure legend.  Some additional explanation should be provided about   how counts/statistics were performed.

Appearance of ubiquitinated PCM1 on western blots:

In Fig 3C, IPs with PCM1 were probed with different ubiquitin antibodies to discern levels and types of chains.  Authors should explain why only single bands appear, suggesting a major monoubiquitinated form.  This is the case not only for FK2, but also for the K63 and K48 linkage antibodies.  Polyubiquitination associated to K63 and K48 chain linkages usually gives rise to smearing up the gel due to slower migrating species.  Additionally, in the upper panel of Fig3E, a construct encoding HA-K63  shows a single band with smearing “down” the  gel.  Authors should explain these results.  Were any blots performed using lower percentage gels with extended running times to resolve high molecular weight species?

Minor note: I assume HA-K63 encodes ubiquitin with all lysines mutated to arginine, except for K63.  If correct, this should be mentioned in M&M.  Also, line 297-298: “K63 and K48 modifications are two common types of ubiquitination and they can be discerned by specific antibodies”: In this section or in Introduction, authors must define better what K48/K63 modifications are.  While this may be understood by ubiquitin experts, many readers will not know that this refers to polyubiquitin chains of different architectures and functions, dependent on which lysines are used.

Linkage-specific ubiquitin chain antibodies and IF:

Fig 3D. Authors make the surprising conclusion that that specific K63 ubiquitin chain linkages can be detected by IF, colocalizing with PCM1.  As K63-linked chains are found throughout the cell, it is quite unlikely that such a specific signal for modified PCM1 could be obtained.  Unless the authors can provide convincing proof and controls for this observation, this data should be removed.

Author Response

Reviewer 2:

Ciliogenesis requires the concerted activity of multiple cellular pathways to construct this important organelle.  Hundreds of proteins are required and coordination of transport and stability is a key mode of regulation.  The manuscript by Wang and colleagues focuses on the ciliogenic role of SNX17, one of a family of sorting nexins that function in protein trafficking within the endosomal network.  They document a striking loss of centriolar satellite markers (including PCM1) upon serum starvation in SNX17 knockout cells.  They further implicate MIB1 as being the responsible E3 ligase for targeting PCM1, but in absence of a physical MIB1-SNX17 interaction, they turned their focus to USP9X, a deubiquitinase known to target centriolar substrates.  They document and map the physical interaction of SNX17 and USP9X, and propose a plausible model that SNX17 helps recruit USP9X to centriolar satellites upon serum starvation to offset MIB1 activity and stabilize PCM1.

Although the initial reason to focus on SNX17 is not stated, this work is interesting and should be published in a revised form, pending clarifications and additional controls.  Although many works have been published on proteolytic control at centrosomes, few have addressed how this control is linked to the vesicular trafficking network, making this work timely and important.

I have the following concerns/suggestions for the authors’ consideration:

SNX17 antibody and localization:

Although a western suggests that SNX17 antibody is specific (Fig 1A; 1E), the antibody is also used in immunofluorescence (Fig 2A).  Authors should validate the specificity of the antibody in IF using the SNX17-KO cells.

Our response: The SNX17 antibody works fine in western blot but not in immunofluorescence staining (Supplementary Table S1). In Figure 2, ectopically expressed SNX17-GFP was used to analysis the localization of SNX17. We are sorry for the confusion. We have revised the Figures (Figure 2B and Figure S4) as well as the main text to clarify this confusion.

Rescue:

Using the SNX17 antibody, authors should provide a western showing levels of SNX17-GFP is the rescue experiments shown in Fig 1F-1G.

Our response: We analyzed the expression level of SNX17-GFP by western blot using the SNX17 antibody and put the result in the revised Figure 1F. The SNX17-GFP protein level in mutant cells is comparable to endogenous SNX17 in WT cells.

SNX17 in fed versus starved cells:

In Fig 2B, authors show that PCM1/CEP131/OFD1 etc are unstable upon cell starvation (also USP9X in Fig 6B).  Although IF images in Fig 2A suggest that SNX17 levels do not change much upon cell starvation, a western should also be provided to support this.

Our response: We now have compared the expression levels of SNX17 in serum-fed and serum-free media by western blot and found serum-starvation did not change the protein level of SNX17 (new Figure 2A).

Overlay images:

Throughout the manuscript, many 3-color overlay images are shown for IF experiments.  It would be useful and recommended to also show grayscale, single-channel images, or channel-separated/zoomed insets (e.g. Fig 6A), especially in cases where co-localization is suggested (e.g. Fig 1F; 2A, 3B, 3D, 4C, 6C).

Our response: We have revised the figures to include channel-separated/zoomed inset for all co-localization experiments as suggested (Figure 2B, 3B, 4B, 5C, 7A, 7C, S2 and S4) to improve the visualization.

Data points/P-values/graphs/tables:

P-values were mentioned several times in text, without supporting data in tabular or graphic form.  Authors should attempt to address this wherever possible.

Specific examples:

Fig 2C. In text, there is mention of p<0.002 in comparing PCM1 in WT versus SNX17-KO cells in starvation. Unless data is provided as a graph and detail given of how data was analyzed/quantitated, the mention of p-value does not seem appropriate.  Counts and raw data can be provided as supplementary or extended data to support the images in Fig 2C.

Line 375-379:  Colocalization of USP9X and PCM1 were detected in WT cells in the presence (84.6%) or absence of serum (84.4%, average of three biological repeats, N>50 in each repeat) (Figure 6A). Meanwhile, this co-localization in mutant cells were reduced from 83.0% in serum-containing media to 9.0% in serum-free media (p<0.0001) (Figure 6A).

Our response: For ciliogenesis assay (Figure 1C, G and Figure 5D), three biological repeats were performed and at least 100 cells were counted for ciliogenesis for each condition. We provided graphs to present mean ± SD and p-value analysis for these experiments. For all co-localization studies (Figure 2B, 3B, 4B, 5C, 7A, 7C, S2 and S4), assays were repeated and total counts (sum of two experiments) were present in the main text/legend in the revised manuscript. We removed p-value analysis for all co-localization studies.

Supp FigS2: TEM analysis and counting analysis of centrosomes, ciliary vesicles and cilia in WT or SNX17-KO cells.  Since few examples were counted, it would be useful to provide more images of centrosomes and ciliary vesicles identified (not just single examples). The text mentions “p=0.0115”, however no analysis is shown in figure or mentioned in figure legend. Some additional explanation should be provided about how counts/statistics were performed.

Our response: TEM analysis were performed twice. We now provide multiple images as well as total counts for WT and MU1 in Supplementary Figure S3 and revised the legend. The p-value in previous submission is derived from unpaired two tailed Student’s t-test for the percentages of CV positive sample. We have now removed this p-value analysis in the revised manuscript.

Appearance of ubiquitinated PCM1 on western blots:

In Fig 3C, IPs with PCM1 were probed with different ubiquitin antibodies to discern levels and types of chains.  Authors should explain why only single bands appear, suggesting a major monoubiquitinated form.  This is the case not only for FK2, but also for the K63 and K48 linkage antibodies.  Polyubiquitination associated to K63 and K48 chain linkages usually gives rise to smearing up the gel due to slower migrating species.  Additionally, in the upper panel of Fig3E, a construct encoding HA-K63 shows a single band with smearing “down” the gel.  Authors should explain these results.  Were any blots performed using lower percentage gels with extended running times to resolve high molecular weight species?

Our response: In our analysis of ubiquitination of PCM1 by FK2, K63 or K48 antibody, we constantly identified one major band (10% PAGE gel with regular running time), indicating a dominant ubiquitination form of PCM1. The exact nature of this modification is not clear, however, it is consistent with a report by Villumsen et al. (PMID: 24121310). In that paper, the authors investigated the MIB1 induced ubiquitination of PCM1 under stress induced ciliogenesis and they observed one major ubiquitinated PCM1 band by western blot (Figure 4C, D, E). Together with our observations, these data indicate that there is one major ubiquitination form of PCM1 during ciliogenesis. The smearing down of K63 blot in original Figure 3E is likely due to partial degradation of the sample. We repeated the experiment and provided image with better quality now (Figure 4D in the revised manuscript). 

Minor note: I assume HA-K63 encodes ubiquitin with all lysines mutated to arginine, except for K63.  If correct, this should be mentioned in M&M. 

Our response: Yes. We have revised the ‘2.1 section of M&M’ to include descriptions of the HA-K63 and K48 ubiquitin mutants used in this study.

Also, line 297-298: “K63 and K48 modifications are two common types of ubiquitination and they can be discerned by specific antibodies”: In this section or in Introduction, authors must define better what K48/K63 modifications are.  While this may be understood by ubiquitin experts, many readers will not know that this refers to polyubiquitin chains of different architectures and functions, dependent on which lysines are used.

Our response: We have revised this section to include a brief introduction to the K48/K63 modifications of protein and two commonly used techniques to investigate the K48/K63 modifications. With these revisions, we believe the result is clearly presented and is accessible to general readers.

Linkage-specific ubiquitin chain antibodies and IF:

Fig 3D. Authors make the surprising conclusion that that specific K63 ubiquitin chain linkages can be detected by IF, colocalizing with PCM1.  As K63-linked chains are found throughout the cell, it is quite unlikely that such a specific signal for modified PCM1 could be obtained.  Unless the authors can provide convincing proof and controls for this observation, this data should be removed.

Our response: We agree with the reviewer’s comments and have removed the original Figure 3D (now Figure 4 in the resubmission).

Reviewer 3 Report

This is a well-executed study demonstrating the role of Snx17 in ciliogenesis. The authors' central claim is that Snx17 regulates cilium formation by recruiting the deubiquitinase Usp9x to PCM1 and therefore counteracting the K63-linked ubiquitination of PCM1 by MIB1. Although the study appears to be technically sound, there are some contradictions with published work and other issues that the authors must address in order to substantiate their claims.

Major issues:

It is not clear to me what the motivation behind this study was. So far as I could tell there were no prior mentions of the role of Snx17 in ciliogenesis, and the authors also don't bring up any particular reason that they became interested in this protein specifically in the context of cilia formation. If the protein was selected based on some kind of screen, the authors should mention it and present the other data from that screen. The finding of K63-linked ubiquitination of PCM1 by MIB1 is not clearly demonstrated. Cajanek et al., 2015 (Journal of Cell Science)  show that MIB1 utilizes K48-linked ubiquitination, and Douanne et al., 2019 (Cell Reports) show that PCM1 is stabilized by MG132, suggesting traditional K48-linked proteasome-targeted ubuiquitination. The authors should cite these two papers and discuss why they are seeing different results. More importantly, the authors should show in Fig. 4A that it is the K63-linked PCM1 that is reduced in MIB1 KD cells (as in Fig. 3C, E). Similarly, Han et al. 2019 (which is cited in the manuscript), show that USP9X knockdown can be counteracted by MG132, suggesting that Snx17 (whose knockdown is not counteracted by MG132) and USP9X act through different mechanisms. The authors should show that USP9X knockdown mimics the effects of Snx17 knockdown/mutation i.e. that it destabilizes PCM1 by increasing its K63-linked ubiquitination and lysosomal targeting (essentially reproduce Fig. 3 but with USP9X knockown rather than Snx17 mutation) In Fig. 5 the authors appear to show that USP9X and PCM1 bind to Snx17 in the same domain. Wouldn't one displace the other if that is the case? The authors should discuss this complication or perform experiments showing this is not the case (for instance by overexpressing one of the proteins and showing that this does not reduce the binding of the other).

Minor issues:

In Fig. 3C.E. the authors should put a label saying "starvation time" next to the numbers above the blot. Otherwise it's not clear what is meant by the numbers When describing results in Fig. 3D the authors claim that it shows PCM1 K63 ubiquitination. The figure merely shows colocalization of K63 with PCM1, but it could well be a different protein in close vicinity to PCM1 that is being ubiquitinated. The authors should tone down their claim in the text and acknowledge this limitation. The authors should describe the method of genotyping of mutant cells in more detail. There seems to be a typo in polybrene concentration (8mg/mL rather than 8μg/mL)

Author Response

Reviewer 3:

This is a well-executed study demonstrating the role of Snx17 in ciliogenesis. The authors' central claim is that Snx17 regulates cilium formation by recruiting the deubiquitinase Usp9x to PCM1 and therefore counteracting the K63-linked ubiquitination of PCM1 by MIB1. Although the study appears to be technically sound, there are some contradictions with published work and other issues that the authors must address in order to substantiate their claims.

Major issues:

It is not clear to me what the motivation behind this study was. So far as I could tell there were no prior mentions of the role of Snx17 in ciliogenesis, and the authors also don't bring up any particular reason that they became interested in this protein specifically in the context of cilia formation. If the protein was selected based on some kind of screen, the authors should mention it and present the other data from that screen.

Our response: We previously studied the role of SNX17 in Notch signaling in zebrafish. We noticed that morpholino knockdown of zebrafish SNX17 induced ciliopathy related phenotype (kidney cyst formation). However, this in vivo study cannot tell whether SNX17 play a direct or indirect role (as reported for the regulation of ciliogenesis by SNX27, PMID: 27974614) in ciliogenesis. So we turn to the well-established RPE1 cell based in vitro ciliogenesis model to investigate the potential function of SNX17 in cilium formation. We now briefly introduce the motivation behind this study in the beginning of the ‘Results’ section.

The finding of K63-linked ubiquitination of PCM1 by MIB1 is not clearly demonstrated. Cajanek et al., 2015 (Journal of Cell Science) show that MIB1 utilizes K48-linked ubiquitination, and Douanne et al., 2019 (Cell Reports) show that PCM1 is stabilized by MG132, suggesting traditional K48-linked proteasome-targeted ubuiquitination. The authors should cite these two papers and discuss why they are seeing different results. More importantly, the authors should show in Fig. 4A that it is the K63-linked PCM1 that is reduced in MIB1 KD cells (as in Fig. 3C, E).

Our response: In addition to induce K48-linked ubiquitination of PLK4 (Cajanek et al. 2015), MIB1 has been reported to induce K63 ubiquitination of substrate such as BLM (PMID: 30044990) and TBK1 (PMID: 21903422). We detected both K63 and K48 ubiquitination of PCM1 (Figure 4C, D), indicating they co-exist in cell and are not mutually exclusive. We demonstrated here that, in SNX17 mutant cells under serum-starvation, more K63 ubiquitinated PCM1 was accumulated than K48 ubiquitinated PCM1 (Figure 4C, D). We found that inhibition of the lysosomal degradation pathway was able to block the degradation of PCM1 which is consistent with the elevated K63 ubiquitination of PCM1 (Figure 4A, B). Now we provide additional data shown that the knockdown of MIB1 by siRNAs reversed the accumulation of K63 linked PCM1 (Figure 5A), further indicating a MIB1 induced K63 ubiquitination and lysosomal degradation pathway in the regulation of PCM1 during serum-starvation induced ciliogenesis in RPE1 cells. Of course, these results do not exclude the possibility of the K48 ubiquitination related proteasome degradation pathway under different conditions or in different cell lines.

In a recent study, Douanne et al. convincingly showed that CYLD regulates the K63 ubiquitination and homeostasis of MIB1. The authors further reported that CYLD deficiency induces degradation of PCM1 which is partially rescued by proteasome inhibitor MG132 (Figure 3F, Cell Rep 2019), indicating a K48 ubiquitination and proteasome degradation pathway for PCM1. However, the authors did not analyze PCM1 ubiquitination directly. The authors did not test the possible function of lysosomal degradation pathway inhibitor such as Bafilomycin A1 in their study, thus their result cannot rule out the possibility that K63 ubiquitination play essential role in the homeostasis of PCM1. We detected both K63 and K48 ubiquitinated PCM1 (Figure 4C and D) and it is likely that they both contribute to the homeostasis of PCM1. However, the enhanced degradation of PCM1 in SNX17 mutant cells during serum-starvation was mainly due to the elevated K63 ubiquitination and the lysosomal degradation pathway. We discussed these papers in the ‘Discussion’ and provided the result of the suggested MIB1 knockdown experiment in Figure 5A.

Similarly, Han et al. 2019 (which is cited in the manuscript), show that USP9X knockdown can be counteracted by MG132, suggesting that Snx17 (whose knockdown is not counteracted by MG132) and USP9X act through different mechanisms. The authors should show that USP9X knockdown mimics the effects of Snx17 knockdown/mutation i.e. that it destabilizes PCM1 by increasing its K63-linked ubiquitination and lysosomal targeting (essentially reproduce Fig. 3 but with USP9X knockdown rather than Snx17 mutation)

Our response: Han et al. recently reported that knockdown of USP9X reduces PCM1 protein level in Hela and HCT116 cells under normal culture condition. The authors further indicated that the degradation of PCM1 is blocked by the proteasomal pathway inhibitor MG132 (this is complicated by the observation that MG132 treatment clearly reduced the protein level of PCM1 in control siRNA treated cells as well, Figure 3C of the paper). In our study, we analyzed the regulation of PCM1 in a different cell line (RPE1) and under different culture condition (serum-starvation) and reported that the lysosomal degradation pathway is important for the homeostasis of PCM1 under serum-starvation. We now performed additional experiments as suggested by the reviewer to further dissect the regulation of USP9X on PCM1. As shown in supplementary Figure S6, we found that siUSP9X treatment of RPE1 in serum media did not induce the degradation of PCM1. However, serum-starvation greatly induced the degradation of PCM1 in USP9X deficiency cells. We then analyzed the ubiquitination of PCM1 and found that the total as well as the K63 linked ubiquitination of PCM1 were enhanced in USP9X knockout cells while K48 ubiquitination of PCM1 was less affected (Supplementary Figure S6). This is consistent with our model that SNX17 recruits USP9X to remove the K63 ubiquitination of PCM1.

In Fig. 5 the authors appear to show that USP9X and PCM1 bind to Snx17 in the same domain. Wouldn't one displace the other if that is the case? The authors should discuss this complication or perform experiments showing this is not the case (for instance by overexpressing one of the proteins and showing that this does not reduce the binding of the other).

Our response: We performed the competition assay suggested by the reviewer by co-transfecting cells with HA-tagged SNX17 (2 µg) and variable amounts of Flag-tagged USP9X (0 to 4 µg) then performed SNX17 pull-down assay. As shown in Supplementary Figure S5, the pull-down of endogenous PCM1 by SNX17-HA seems not affected by increasing amount of USP9X, indicating that PCM1 and USP9X do not compete with each other for binding to SNX17.

Minor issues:

In Fig. 3C.E. the authors should put a label saying "starvation time" next to the numbers above the blot. Otherwise it's not clear what is meant by the numbers.

Our response: We have revised those Figures (Figure 4C, 4D, S6 in the revised manuscript) with the label ‘starvation time’ as suggested.   

When describing results in Fig. 3D the authors claim that it shows PCM1 K63 ubiquitination. The figure merely shows colocalization of K63 with PCM1, but it could well be a different protein in close vicinity to PCM1 that is being ubiquitinated. The authors should tone down their claim in the text and acknowledge this limitation. The authors should describe the method of genotyping of mutant cells in more detail. There seems to be a typo in polybrene concentration (8mg/mL rather than 8μg/mL)

Our response: We agree with the reviewer’s comment and removed the co-localization data as suggested by reviewer #2 and revised main text accordingly. We described the method of genotyping of mutant cells in the revised ‘M&M” (section 2.3. SNX17 knockout). The polybrene concentration we used is indeed 8 µg/ml as described in the original papers (PMID:     6301409; PMID: 9797877).

Round 2

Reviewer 1 Report

The authors have addressed my previous concerns. I have no further comments.

Reviewer 3 Report

The authors have addressed all my concerns and have significantly strengthened the manuscript. I recommend publication without further revisions. Congratulations on an exciting study!